# Effect of the FIFA 11+ soccer specific warm up programme on the incidence of injuries: A cluster-randomised controlled trial

**Assuman Nuhu**[1,2][◉] *, **Jennifer Jelsma**[2][◉], **Kim Dunleavy**[3][‡], **Theresa Burgess**[2][◉]

**1** Department of Physiotherapy, College of Medicine and Health Sciences, University of Rwanda, Kigali, Rwanda, **2** Division of Physiotherapy, Department of Health and Rehabilitation Sciences, University of Cape Town, Groote Schuur Hospital, Cape Town, South Africa, **3** Department of Physical Therapy, University of Florida, Gainesville, FL, United States of America

◉ These authors contributed equally to this work.
‡ These author also contributed equally to this work.
* nuhu.assuman@gmail.com

## Abstract

### Background

Soccer players incur injuries that typically affect their performance. Injuries are caused by intrinsic and extrinsic factors that call for multifactorial preventive interventions. The study examines the impact of the FIFA 11+ warm up programme on the incidence and severity of injuries in second division soccer players in Rwanda.

### Methods

Twelve teams (309 players) were randomised in the intervention group and 12 teams (317 players) in the control group using a cluster randomized controlled trial with teams as the unit of randomization. Intervention group teams implemented the FIFA 11+ soccer specific warm-up programme during training and matches at least three times a week over seven months of the Rwandan soccer season. Control group teams continued with usual warm up exercises. The primary outcome of this study was the overall incidence of training and match injuries. Injuries, training and match exposure as well as severity categories were recorded per the F-MARC guidelines.

### Results

A lower proportion of players sustained injuries in the intervention group (52%) compared to the control group (63%) (Odd ratio: 0.7; 95%CI: 0.5–0.9). A significantly lower rate ratio was observed in the intervention group for overall (RR = 0.6; 95%CI: 0.5–0.8) and match (RR = 0.6; 95%CI: 0.5–0.8) injuries. Compliance to the injury prevention programme was 77%. In the intervention group, the incidence of injury was similar across all teams and across the medium and highly compliant teams. There was a statistically significant 55% and 71% reduction of the rate of moderate and severe injuries in the intervention group respectively.

**Data Availability Statement:** All relevant data are within the manuscript and its Supporting Information files.

**Funding:** The study was funded by the Government of Rwanda though a PhD scholarship by the Rwanda Education Board. The funder had no role in study design, data collection and analysis, decision to publish, or preparation of the manuscript.

**Competing interests:** The authors have declared that no competing interests exist.

## Conclusion

The 11+ warm up injury prevention programme resulted in a significant reduction in the odds of sustaining injuries. In addition, injuries sustained were less severe. The programme should be rolled out to all teams in Rwanda and may well result in a decrease in the incidence and severity of injury in similar contexts.

## Trial registration

Pan African Clinical Trial Registry (PACTR201505001045388).

## Introduction

The incidence of injury in soccer players is high at all levels and ranges from 1.2 to 18.8 per 1000 hours [1–3]. For individual soccer players, the loss of income due to injury may have severe consequences as alternative employment possibilities may be decreased, particularly in low-income countries [4]. Effective injury prevention is therefore an important goal for professional teams. Injuries generally occur more frequently during matches than during training and most commonly in the lower extremities [5–7]. Sprains, strains and contusions are the most prominent types of injuries [8–10] resulting in extended periods of absence from training and match participation and lead to medical and rehabilitation expenses [11, 12]. Loss of key players due to injuries may also have an impact on the performance and success of teams [13]. There is therefore a constant need for injury prevention studies to identify the effectiveness of prevention programs on actual incidence of injury using prospective methods.

Studies have examined the effects of several interventions in preventing injuries in soccer using one or more strategies [14–20]. An orthosis have been reported to be supportive after previous ankle injuries but no effect was found for initial injuries [15, 16]. There is also good evidence supporting balance and proprioceptive exercises for preventing recurrent ankle sprains [15, 17, 18] and ACL injuries [19]. In contrast, stretching and flexibility exercises did not reduce the risk of muscle and tendon injuries [20, 21]. Similarly video-based awareness interventions did not have any effect in the reduction of the number of soccer injuries [22]. Reliable evidence is available supporting the impact of Nordic eccentric hamstring exercises to prevent hamstring injuries [20, 23–25], nevertheless strengthening exercises were not found to be effective in preventing other soccer injuries [26].

Due to the complexity of the game and the multi-component aetiology of injuries, multi-modal interventions have been recommended [27]; programmes incorporating core strength, balance, coordination and plyometrics have been reported to reduce overall injuries [28, 29]. Encouraged by these studies, the Federation of International Football Associations (FIFA) developed a multi-component soccer specific injury prevention intervention programme [30]. The FIFA 11+ injury prevention programme was developed by the group of experts commissioned by FIFA to reduce perceived intrinsic risk factors for injuries in soccer [30]. The programme is composed of 15 evidence-based exercises targeting strength, balance and coordination performed in sequence during warm up. The three components involve: 1) slow running and dynamic stretching, 2) six sets of exercises targeting the trunk and legs with three progression levels and 3) running drills with increased speed with different complexity of movements. The exercises are recommended to be performed at least three times per week [30]. The FIFA 11+ programme increased general lower limb strength as measured by

countermovement and squat jumps [31], increased peak torque of concentric and eccentric hamstring strength [32], leading to improvement of neuromuscular control and the correct alignment of the hip, knee and ankle joints in joints during dynamic activities in young athletes [33]. Therefore the programme may play an important role in preventing lower limb injuries [18, 21, 24–26, 34–36].

At the time of this study, the positive impact of the FIFA 11+ warm up in preventing injuries had been studied with only 4 randomized controlled trials (RCT) available [37–40]. Since this time, others have concluded that the FIFA 11+ programme is effective for soccer players of different ages and levels [41–43]. Among the four RCTs that assessed the effect of the FIFA 11+ programme on the occurrence of injuries, three were conducted among male soccer players. Two of them were conducted among youth [38, 39] while one was conducted among veterans [40]. In Nigeria, a study randomised 20 youth teams into an intervention group that performed the FIFA 11+ warm-up programme (10 teams with 212 players) and a control group instructed to continue with their standard warm-up (10 teams with 204 players) [38]. A significant reduction in incidence of all injuries and lower extremity injuries was observed in the intervention group compared to the control group. However, the study did not find any significant difference in specific injuries such as in the thigh, knee, and ankle. Further, the study did not have any effect on the severity of injuries and did not specifically impact incidence of contact or non-contact injuries [38]. Another RCT of 1525 collegiate male soccer players from the USA demonstrated the protective effects of the FIFA 11+ in reducing injuries [39]. The intervention group players (n = 675) were instructed to perform the intervention three times per week for the full season. The control group players (n = 850) used their standard warm-up routine. The intervention reduced overall injuries by 46% and significantly reduced time loss due to injuries. Although the study shows the effectiveness of the warm- up programme, collegiate level teams had less exposure instances as they only participated in 18 matches and completed 51 practice sessions [39]. In addition, the collegiate-level season was only five months in length, which is shorter compared to most professional seasons lasting eight to ten months [39].

The two studies conducted in youth show that the intervention could reduce the incidence of overall, match [38] and training [39] injuries. Though the studies found a reduction in lower limb injuries, no significant difference was observed in the incidence of specific regional injuries in the thigh, knee and ankle regions. One study reported a reduction in time lost due to injuries while the other one did not find any effect on the severity of injuries.

It was therefore important to examine the intervention programme in a different population. The Rwandan second division youth soccer league is an amateur newly established league with relatively limited competitive experience, potentially predisposing players to more injuries and the FIFA 11+ had not been used in Rwanda. The training context and implementation of the FIFA 11+ programme in a population with limited experience with the warm up and relatively limited intensive soccer experience was of interest. Therefore the aim of this study was to determine the impact of the FIFA 11+ warm up programme on incidence, location, and severity of injuries in amateur second division soccer players in Rwanda.

## Methods

### Design and ethical approval

A cluster-randomized controlled design was used in this study. The study was approved by the Faculty of Health Sciences Human Research Ethics Committee from the University of Cape Town (HREC/REF: 683/2014) and registered in the Pan African Clinical Trial Registry (PACTR201505001045388). The Federation Rwandaise de Football Associations (FERWAFA)

and the clubs gave permission to conduct the study while coaches and players (parents or guardians of the under-18 players) signed consent forms indicating their agreement to participate in the study. Teams were randomized rather than players as implementation of the programme was at the team level.

## Participants

Soccer players from all the 24 Rwandan Second Division teams registered in the 2016 season were invited individually to participate in the study. There were 630 players who were eligible for recruitment in their respective teams. Teams registered a mean of 26 (SD = 3) players, ranging between 21 and 37 players per team. The mean age of the players was 20 (SD = 1) years old. Players were relatively inexperienced with a mean of three years playing soccer, while the most experienced player had only six years playing experience. They reported a mean of 2 (SD 1) years in the second division and the same experience in their current team of 2 (SD 1) years. The mean body mass of the players was 65 (SD = 4) kg, mean stature was 169 (SD = 4) cm and mean body mass index (BMI) of the players was 22.6 (SD = 0.9) kg.m$^{-2}$.

## Sample size calculation

Sample size calculation was based on the prevalence study conducted by Twizere [44] who reported that approximately 70% of soccer players in Rwanda sustained injuries in a year. Using this estimate and targeting a 20% reduction of injuries using power of 90% for generalised linear regression model and two-tailed alpha of 5%, a sample size of 125 was required per group. Taking into consideration the design effect of two for cluster randomization, the minimum number of players needed was 217 in each group. The mean number of 20 players per team would be sufficient given the intra-class correlation of 0.03 [45].

## Randomisation and allocation of the participants

Random sampling was used to allocate the 24 teams to the study groups. The numbers representing the the 24 teams were placed in sealed envelopes with an equal opportunity of being selected for groups. A research assistant selected the envelope for both groups drawing one by one to achieverandom allocation by team to either the intervention group or the control group. The principal investigator, assisted by the Rwanda soccer technical director, supervised the randomization process.

## Inclusion criteria

Players who had contracts and licenses from the teams enrolled to play in the second division were included in the study. Players with pre-existing injury at the beginning of the season were included, however data related to the pre-existing injuries were not included in data analysis. Injury and exposure data were obtained for all the participants.

## Instruments

**Four forms were used to collect data.**

1. Demographic data: A self-designed form was developed to gather demographic data and information pertaining to the coaching or playing experience.

2. Training exposure registration form: Coaches were provided a standardised exposure form developed by FIFA[2] to record detailed information related to duration of training and player involvement. Any form of physical activities supervised by the coach was recorded as

training exposure time in minutes. The duration of the training was recorded from the start of the warm up to the end of the cool down session [46]. The research team collected the forms every two weeks.

3. Match exposure form: Match exposure was gathered from the referees' reports submitted to FERWAFA. Match exposure time was defined as the amount of time the player was involved while the team played against another team.

4. Injury report form: An injury report form was completed by the team medical personnel to collect data pertaining to the incidence of injuries [47]. Definitions of the type, anatomical location, severity and circumstance of injury were provided at the back page of the form. The form was developed and used during major FIFA tournaments and Olympic games [48] and has been extensively used in other studies [49]. For example an injury is defined as any physical complaint that resulted from soccer participation whether in training or match. It does not necessarily mean that there is a need of medical attention. The severity of injury was classified according to the number of days of absence from full soccer participation due to injury. Injuries that resulted in an absence from training or matches for 1–3, 4–7, 8–21 and more than 21 days were classified as "minimal", "mild", "moderate" and "severe" injuries respectively [50].

## Intervention

**Control group.** Coaches of the teams in the control group were invited to attend a training on basic first aid and emergency procedures for one day. The coaches received training from the researcher on basic first aid and emergency procedures. The training covered topics relating to the occurrence of injuries, field assessments, and phases of rehabilitation. First aid for injuries affecting specific body parts (ankle, knee, hip, groin, hamstring, hand and head) including concussion were also discussed. The discussion focused on specific conditions that affect players on the field such as nosebleeds, eye and dental injuries. They were requested to continue with the way they usually organise the warm up and practices for their players. There was no specific warm up regime in the control group. None of the control group coaches were familiar with the FIFA 11+ warm up programme. Typically the soccer training started with a warm up, followed by technical and/or tactical drills. However, depending on the period of the season physical training may include specific exercises for cardiovascular endurance, muscle strength, coordination and speed for performance.

**Experimental group.** *Procedure*. During the pre-season period, coaches, players and medical personnel of all the second division teams in the experimental group were invited to attend an instructional meeting to present the purpose of the study to clarify roles. An instructor recognized by the Confederation Africaine de Football (CAF) introduced the concepts of injury prevention, the development, and use of the FIFA 11+ injury prevention programme. The FIFA 11+ involves slow speed runs combined with dynamic stretches; followed by exercises designed to strengthen the trunk muscles and balance exercises targeting the lower limbs; and finally running at an increased speed including different drills. A power point presentation and video were used to describe each exercise and in-depth explanations of the appropriate movements and positions were presented (S1 Appendix). Coaches also took part in practical sessions performing each exercise while the instructor provided corrections for flaws in technique. On the third day of the training, the coaches played the roles of both the instructor and the player. Coaches were provided with copies of the FIFA 11+ manual and cards as well as a DVD for further reference. Coaches were requested to organize the FIFA11+ programme as a warm up for the players in their teams at least three times a week. The research team collected completed training exposure and injury forms every two weeks.

## Compliance and follow up

The instructor and the principal investigator visited the intervention group coaches once a week for the first month of the season, followed by monthly visits to foster implementation and compliance. Additional instructions were provided where necessary. The research assistants collected the completed forms twice a month. The medical personnel recording the injuries and research assistants were blinded to the group.

## Statistical analyses

Descriptive statistics were used to present baseline characteristics. A weighted independent t-test was used to assess the difference in the demographic characteristics of the players between the two groups. The primary outcome with respect to efficacy of the FIFA 11+ warm up programme was the comparison of incidence of overall, training and match injury per 1000 hours of exposure with a 95% confidence interval between the intervention and control groups. Comparison of injury rates (Rate Ratio, RR) by study group was based on a team-adjusted Poisson regression analysis fitting the generalized linear regression model. Compliance was computed based on each team's implementation of the FIFA 11+ sessions throughout the season as a percentage of total training and match sessions and categorized as high compliance (> 75%), medium compliance (50% to 74%), or low compliance (<50%). The influence of compliance on the injury outcome was also analysed. Additional analyses were conducted for region of injury, lower limb injury location, and the severity of injuries. The categories of the severity of injuries were reflected by days of absence from full soccer participation [50]. Microsoft Excel (Version 2000, Microsoft, Redmond, Wash.) was used to capture the data. Data analysis was carried out using "Statistica" data analysis software, version 13.2, Dell Inc. and the Statistical Package for Social Sciences (SPSS) for Windows, version 21.0, SPSS Inc, Chicago, USA.

## Results

Of the 25 teams that participated in the 2016 second division league, 24 (630 players) agreed to participate in the study and were randomised (intervention 12 teams, 312 players; control 12 teams, 318) (Fig 1). Five players (4 intervention, 1 control) were lost to follow up. No player was injured before the start of the season.

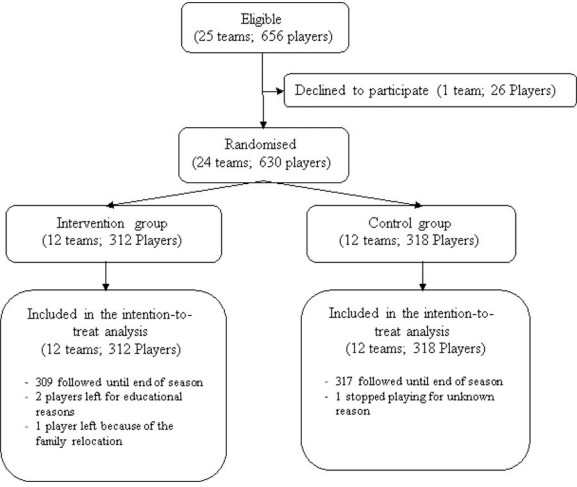

**Fig 1. Flow chart of participants through trial.**

## Characteristics of players in the intervention and control group

The independent t-test was conducted after assessing the normality of data and adjusting for clustered teams. There were no statistically significant differences in the demographic characteristics of the players from the intervention group or control groups (Table 1).

## Training and match exposures

Players in the intervention group completed 61043 hours of training and 4290 match hours, with 65333 total exposure hours in the season. The control group players were exposed to 59165 hours of training and 4224 match hours, totaling 63389 hours. Teams in the intervention group had significantly greater total overall and training exposure hours than teams in the control group (p<0.001). however, there was no significant difference in match exposure between the two groups (p = 0.367).

## Compliance with the injury prevention programme and injuries

The teams in the intervention group implemented the injury prevention programme in approximately 75% of combined training sessions and matches (range 60–97%, mean 3.6 times/week, range 2.8–4.9/ week). Teams in the intervention group demonstrated moderate to high compliance. No difference was found in either the overall injuries or any other subtype of injuries according to the levels of compliance.

## Effects of the FIFA 11+ injury prevention intervention injury incidence

There were 420 time loss injuries (intervention n = 168,40%, control n = 252,60%). Taking exposure into consideration, the incidence of injuries in the intervention group was 2.6 (95% CI: 2.2–3.0) injuries/ 1000 hours of exposure and 4.0 (95%CI: 3.5–4.5) injuries/ 1000 hours of exposure in the control group. Statistically significant lower incidence was observed in the intervention group compared to the control group for overall (p<0.001), training (p<0.05) and match (p<0.001) injuries (Table 2).

## Effect of the FIFA 11+ injury prevention on injury risk

Out of a total 312 players in the intervention group, 163 (52%) players sustained injuries while in the control group, 200 out of 318 (63%) players incurred injuries. Of those players who sustained injuries, the majority sustained only a single injury. Although the frequency of players sustaining one injury in either group was similar (109 in the intervention group compared to 98 in the control group), the control group had more than twice the number of multiple injuries per player over the season. The odds of sustaining injuries were significantly lower in the

**Table 1. Demographic characteristics of players in the intervention and control groups.**

| Characteristics | Intervention Group | Control Group | p-value |
|---|---|---|---|
| | Mean (SD) | Mean (SD) | |
| Age (years) | 19.9 (1.5) | 19.7 (1.4) | 0.089 |
| Soccer experience (years) | 2.9 (0.8) | 3.0 (0.9) | 0.328 |
| Second division experience (years) | 1.7 (0.7) | 1.9 (1.0) | 0.131 |
| Current club experience (years) | 1.5 (0.7) | 1.7 (1.0) | 0.223 |
| Weight (kg) | 65.3 (3.5) | 64.4 (4.6) | 0.206 |
| Height (cm) | 169.5 (3.5) | 169.2 (4.0) | 0.486 |
| Body Mass Index (kg.m$^{-2}$) | 22.7 (0.8) | 22.5 (1.0) | 0.188 |

**Table 2. Effect of the intervention on the incidence of overall, training and match injuries.**

| Injuries | Study group | Number of injuries | Exposure (hours) | Incidence (95%CI) | Rate ratio (95% CI) | p-value |
|----------|-------------|--------------------|--------------------|--------------------|----------------------|---------|
| Overall | Intervention | 168 | 65333 | 2.6 (2.2–3.0) | 0.6 (0.5–0.8) | *<0.001* |
| | Control | 252 | 63389 | 4.0 (3.5–4.5) | | |
| Training | Intervention | 64 | 61043 | 1.0 (0.8–1.3) | 0.7 (0.5–0.9) | *0.028* |
| | Control | 89 | 59165 | 1.5 (1.2–1.9) | | |
| Match | Intervention | 104 | 4290 | 24.2 (20.0–29.3) | 0.6 (0.5–0.8) | *<0.001* |
| | Control | 163 | 4224 | 38.6 (33.2–44.8) | | |

intervention group for overall (OR: 0.65; 95%CI: 0.47–0.89) and match (OR: 0.71; 95%CI: 0.50–0.99) injuries. The reduction in the risk of training injuries did not reach significance in the intervention group (OR: 0.73; 95%CI: 0.50–1.09). There were 116 (37%) players that sustained lower limb injuries in the intervention group and 144 (45%) in the control group. The risk of sustaining lower extremity injuries was significantly lower in the intervention group (OR: 0.72; 95%CI: 0.52–0.98), with a strong trend for the reduction in the risk of ankle injuries (OR: 0.52; 95%CI: 0.32–0.86). The odds of contact and non-contact injuries were also significantly lower in the intervention group compared to the control group with a significantly lower risk of traumatic injuries observed in the intervention group (OR: 0.70; 95%CI: 0.51–0.96) (Table 3).

## Effect of the FIFA 11+ injury prevention intervention on injury severity

Players in the intervention group lost 40% less days due to injuries (median 6, interquartile range, IQR: 6; range 1–84 days) compared to the control group (median 8, IQR = 11; range 1–153 days) (p < 0.001). There was a statistically significant reduction of the rate of moderate (55%) and severe (71%) injuries in the intervention group respectively (Table 4).

## Discussion

To our knowledge, this is the second RCT examining the effect of the FIFA 11+ injury prevention programme among male amateur soccer players in an African population. The study

**Table 3. Effect of the intervention on the risk of injuries between study groups.**

| | Intervention group (n = 312) | Control group (n = 318) | Odds ratio (95%CI) | p-value |
|---|---|---|---|---|
| **Injuries** | **n (%)** | **n (%)** | | |
| Overall | 163 (52) | 200 (63) | 0.65 (0.47–0.89) | 0.009 |
| Training | 57 (18) | 74 (23) | 0.73 (0.50–1.09) | 0.122 |
| Match | 85 (27) | 110 (35) | 0.71 (0.50–0.99) | 0.046 |
| **Location of injury** | | | | |
| Lower extremities | 116 (37) | 144 (45) | 0.72 (0.52–0.98) | 0.047 |
| Thigh & groin | 20 (6) | 33 (10) | 0.65 (0.37–1.17) | 0.194 |
| Knee | 38 (12) | 43 (14) | 0.89 (0.56–1.42) | 0.701 |
| Ankle | 27 (9) | 49 (15) | 0.52 (0.32–0.86) | 0.013 |
| **Mechanism of injury** | | | | |
| Contact | 126 (40) | 172 (54) | 0.56 (0.42–0.79) | 0.001 |
| Non-contact | 42 (13) | 80 (25) | 0.46 (0.31–0.70) | <0.001 |
| **Aetiology of injury** | | | | |
| Overuse | 8 (2.5) | 15 (4.7) | 0.53 (0.22–1.27) | 0.219 |
| Trauma | 160 (51) | 191 (60) | 0.70 (0.51–0.96) | 0.033 |

**Table 4. Effectiveness of the intervention on the severity of injuries.**

| Severity of injury | Intervention group (n = 168) | | Control group (n = 252) | | Rate ratio (95%CI) | p-value |
|---|---|---|---|---|---|---|
| | n (%) | Incidence (95%CI) | n (%) | Incidence (95%CI) | | |
| Minimal (1–3 days) | 44 (26) | 0.7 (0.5–0.9) | 40 (16) | 0.6 (0.5–0.9) | 1.07 (0.70–1.64) | 0.850 |
| Mild (4–7 days) | 63 (38) | 1.0 (0.8–1.2) | 69 (27) | 1.1 (0.9–1.4) | 0.89 (0.63–1.25) | 0.543 |
| Moderate (8–28 days) | 52 (31) | 0.8 (0.6–1.0) | 113 (45) | 1.8 (1.5–2.1) | 0.45 (0.32–0.62) | <0.001 |
| Severe (>28 days) | 9 (5) | 0.1 (0.1–0.3) | 30 (12) | 0.5 (0.3–0.7) | 0.29 (0.14–0.61) | 0.001 |

population had relatively limited overall soccer experience and the FIFA 11+ was not used prior to this study, providing a unique context for studying the effect of the warm-up programme. This study showed that the introduction of the FIFA 11+ programme with moderate to high compliance in training and competition decreased the incidence of injuries compared to a control group using their usual warm up. Along with significantly lower overall, training and match incidence rates, the severity of injuries was lower in the intervention group and fewer days were lost due to injury. The programme was implemented in at least three quarters of all sessions, rising to 91% prior to matches. The levels of compliance did not have any impact in the incidence of injuries in the control group. Of importance there were no injuries or adverse side effects while performing the 11+ programme. Only one previous study reported on the side effects of the intervention, as players reported muscle soreness and one hamstring strain during the 11+ warm up programme [39].

The recruitment of all eligible players in the second division and the follow-up over a full season implies that these results might be generalisable to amateur youth players at a similar level of soccer participation. In addition, the satisfactory level of team compliance achieved in the present study provides support for feasibility of implementation. Strengths of this study included adequate sample size, longitudinal data collection andhigh compliance in the intervention group, allowing conclusions on impact of the warm-up. The randomization supports internal validity as there were no significant differences in the demographic variables of the players between the two study groups. The differences in the training exposure could be an external factor influencing the incidence of training injuries especially in the intervention group. The players in the intervention group had significantly higher training exposure hours due to two teams reaching the final stage of the competition with extra training exposure.

There may be several factors contributing to the high compliance levels and ultimately a reduction in injuries. The length of the training was three days, in contrast to previous studies where the duration of training ranged between 1.5 hours [51] and 8 hours [52, 53]. The longer training programme allowed for both theoretical and practical training within a real world context, and enabled coaches to assimilate and master the correct instruction of exercises. Emphasis was placed on the appropriate instructions and corrections of body alignment and execution of the movements. In addition, follow-up visits were made to the teams and contact was maintained with coaches throughout the season.

The motivation of the coaches to improve their practice is an important element of successful training [54]. The use of accredited FIFA training material and careful instruction regarding the correct application of the exercises may have provided the coaches with confidence in the potential of the programme to reduce injuries. A number of studies provided instructional videos to coaches, together with training manuals, posters [27, 55, 56] or a book describing the exercises [52]. Similarly, coaches were provided with DVDs that included descriptions of the exercises, a complete programme of instructions, a booklet and the FIFA link for references. It is possible that the limited training opportunities in the low- to middle-income country setting

contributed to the perceived importance of the opportunity and to the full implementation of the acquired skills.

## Impact of the FIFA 11+ programme on injuries

The overall, training and match incidence rates were all significantly lower in the intervention group, with an 11% reduction in the overall number of injured players and a mean reduction of time off play by six days, compared to the control group.

The intervention significantly decreased the risk of injuries in the intervention group by 35%; and the odds of sustaining injury in the intervention group were comparatively lower (27% and 29% for training and matches respectively). This reinforces the results from previous studies that demonstrated the efficacy of the FIFA 11+ warm up programme in preventing injuries in general [38–40]. A similar positive impact of the FIFA11+ programme on overall or training incidence was reported in two of the three studies that tested the intervention using RCTs [38, 39]. The impact in terms of absolute reduction varied from -11% (current study) to -49% [39] with substantial variability in incidence of injuries among studies. Only one previous study had a population containing the age bracket of the present study but had a greater sample size [39].

In the present study, the warm-up intervention resulted in significantly lower incidence of match injuries. Any reduction in match injuries would be considered beneficial, as match injury incidences are consistently higher than those sustained during training [1, 3, 6, 7, 57, 58]. The intervention resulted in less contact injuries, that can potentially be attributed to the neuromuscular effects of the FIFA 11+ programme. In other studies, the completion of the FIFA 11+ programme twice per week improved the general physical fitness of young players participating in indoor soccer [59], whereas performing the programme three times per week improved neuromuscular control in amateur soccer players in Italy [33]. While it is not possible to make definitive conclusions from the data collection in this study, improvements in neuromuscular control and fitness may have prepared players to withstand contacts that would otherwise have resulted in injuries. However, we were not able to record the types and mechanisms of contact injuries that could be prevented by the programme as contact injuries are often multifactorial. The results of this study are in contrast with the studies conducted by Owoeye et al. [38] and Hammes et al. [34], where the FIFA 11+ programme did not impact contact injuries. However, these studies [34, 38] reported poor compliance rates and it is possible that the neuromuscular benefits of the programme may not have been achieved. The same studies [34, 38] did not find significant reductions in training injury occurrences [34, 38]. Further studies would be important to investigate the level of compliance required to prevent training injuries.

Significantly lower incidences were observed in the intervention group for moderate and severe injuries, which is in accordance with the studies conducted by Silvers-Granelli et al. [39] and Hammes et al. [34]. Reducing severe injuries has a major impact on the total time lost due to injuries. Not only were the players in the intervention group less likely to suffer severe injury, injured players in the intervention group returned to play sooner than players in the control group.

The intervention significantly reduced the incidence of lower extremity and ankle injuries, comparable with the other studies [38, 39]. This is in line with the aspiration of the FIFA Medical Assessment and Research Centre that designed a programme targeting the reduction of lower limb injuries by including exercises to improve neuromuscular control with good posture and correct alignment of lower limb joints [37]. An experimental study conducted in 20 young athletes found that performing the FIFA 11+ warm up programme over a nine-week

period resulted in increased general lower limb strength as measured by countermovement and squat jumps [31]. Analysing muscle activation, Nakase et al. [60] found an increased core muscle activation, specifically the rectus abdominis and the gluteus medius and minimus when performing the FIFA 11+ exercises. Additionally, the rectus abdominis and hip adductor muscles were more active when performing part two of the programme [60]. The increase in abdominal, hip extensor and adductor neuromuscular control may assist with force distribution and proximal control while supporting optimal biomechanical alignment of the hip, knee and ankle joints [33]. However, we were not able to measure changes in specific intrinsic physical factors. We were therefore not able assess the contribution of intrinsic factors in preventing injuries.

The results of the present study complements the existing body of knowledge by providing evidence of the positive effects of the FIFA 11+ prevention programme. The increase in dynamic balance suitable soccer skill performance could lead to the physiological preparedness of the players that subsequently reduce the risk of injuries [61]. It is therefore important to investigate the possibility of extending the use of the preventive programmes to the wider soccer community in Rwanda.

## Compliance with the programme

The mean team compliance level in the present study was 77% for all training sessions and matches. The compliance level is somewhat similar to previous studies ranging between 30% [39] and 77% [62]. Therefore, the recommended implementation rate of twice per week [63] was not only met, but exceeded with teams using the mean of 3.6 times the programme, and a minimum of 2.8 times per week. The high compliance rate may be partially attributed to the motivation levels of the coaches observed during training and follow up. Moreover, the training stimuli comprised of various didactic materials, and the extended practical sessions and regular team visits may have contributed to the conformity to the programme. Coaches may have held beliefs that performing the programme as many times as possible would provide a greater increase in neuromuscular strength increased their motivation. However, further investigation, and perhaps qualitative studies are needed to obtain an in-depth understanding of coaches' motivations for compliance with the injury prevention programme.

## Limitations of the study

The nature of the study required close monitoring and supervision of the coaches for compliance and therefore, it was not possible to blind the researcher or coaches to the group allocations. However, the medical personnel and research assistants were blinded as to which team was allocated to the intervention or control groups as well as nature, type and organisation of the intervention, from training to follow-up.

Further, the study relied on the injury reports from the medical personnel of the teams using subjective and physical examination due to limited funding. It would be interesting to explore barriers and facilitators to the use of standardised injury reporting methods within this context; and to identify how acceptability of injury reporting methods and training loads could be improved.

Future studies may benefit from investigation of individual player compliance. Prior to this study there was no systematic recording of injuries and exposure in Rwanda, therefore recording this type of data was also new to both the coaches and medical personnel. This process placed an additional load to their usual work activities. As a result, coaches were not willing to provide player compliance information as this was too time consuming. Further investigation of the perception of players and coaches could be performed for in-depth analysis of the

awareness and opinion about injury prevention. More studies are also needed to assess the content of the usual warm up done by teams in the control group for further comparison.

## Conclusion and recommendations

This study is the fourth RCT examining the effect of the FIFA 11+ injury prevention programme among male soccer players and the second conducted in an African population. The teams allocated to the intervention group completed the injury prevention programme with the mean of 77% of all training sessions and matches and a mean of 3.6 times per week indicating an acceptable level of compliance with the programme. The intention-to-treat results indicated that the FIFA 11+ programme significantly decreased the incidence of overall, training and match injuries, more specifically lower extremity injury among second division soccer players. Positive effects of the programme on contact and severe injuries are encouraging for reducing serious soccer injuries with consistent implementation. There is therefore a need to incorporate the FIFA 11+ programme in the wider soccer community in Rwanda and similar settings to reduce the burden of injuries.

## Supporting information

**S1 Checklist. CONSORT checklist.**
(PDF)

**S1 Protocol. Study protocol.**
(PDF)

**S1 File. Dataset.**
(XLSX)

**S1 Appendix. Appendix.**
(DOCX)

## Acknowledgments

We thank all coaches and players who participated in this study. Thanks to all the medical staff and research assistants that supported data collection. We also acknowledge the support from the national and international soccer governing bodies.

## Author Contributions

**Conceptualization:** Assuman Nuhu, Jennifer Jelsma, Theresa Burgess.

**Data curation:** Assuman Nuhu, Jennifer Jelsma.

**Formal analysis:** Assuman Nuhu, Jennifer Jelsma, Theresa Burgess.

**Funding acquisition:** Assuman Nuhu.

**Investigation:** Assuman Nuhu.

**Methodology:** Assuman Nuhu, Jennifer Jelsma, Theresa Burgess.

**Project administration:** Assuman Nuhu, Theresa Burgess.

**Resources:** Assuman Nuhu.

**Software:** Theresa Burgess.

**Supervision:** Jennifer Jelsma, Theresa Burgess.

**Validation:** Jennifer Jelsma, Theresa Burgess.

**Visualization:** Assuman Nuhu, Theresa Burgess.

**Writing – original draft:** Assuman Nuhu, Kim Dunleavy.

**Writing – review & editing:** Assuman Nuhu, Jennifer Jelsma, Kim Dunleavy, Theresa Burgess.

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
