## [Decision Letter · Decision Letter 0]

18 Nov 2020

PONE-D-20-09354

Effect of the FIFA 11+ soccer specific warm up programme on the reduction of injuries: a cluster-randomised controlled Trial

PLOS ONE

Dear Dr. Nuhu,

Thank you for submitting your manuscript to PLOS ONE. After careful consideration, we feel that it has merit but does not fully meet PLOS ONE’s publication criteria as it currently stands. Therefore, we invite you to submit a revised version of the manuscript that addresses the points raised during the review process.

The manuscript has been evaluated by several reviewers, and their comments are available below.

The reviewers have raised a number of concerns that need attention. They request additional information on methodological aspects of the study, and revisions to the statistical analyses. 

Could you please revise the manuscript to carefully address the concerns raised?

We look forward to receiving your revised manuscript.

Kind regards,

Carmen Melatti

Associate Editor

PLOS ONE

Journal Requirements:

2. Please include captions for your Supporting Information files at the end of your manuscript, and update any in-text citations to match accordingly. Please see our Supporting Information guidelines for more information: http://journals.plos.org/plosone/s/supporting-information

Reviewers' comments:

Reviewer's Responses to Questions

**Comments to the Author**

1. Is the manuscript technically sound, and do the data support the conclusions?

Reviewer #1: Yes

Reviewer #2: Yes

Reviewer #3: Yes

Reviewer #4: Yes

Reviewer #5: Yes

Reviewer #6: Partly

2. Has the statistical analysis been performed appropriately and rigorously? 

Reviewer #1: Yes

Reviewer #2: Yes

Reviewer #3: No

Reviewer #4: Yes

Reviewer #5: N/A

Reviewer #6: Yes

3. Have the authors made all data underlying the findings in their manuscript fully available?

Reviewer #1: Yes

Reviewer #2: Yes

Reviewer #3: Yes

Reviewer #4: Yes

Reviewer #5: Yes

Reviewer #6: Yes

4. Is the manuscript presented in an intelligible fashion and written in standard English?

Reviewer #1: Yes

Reviewer #2: Yes

Reviewer #3: Yes

Reviewer #4: Yes

Reviewer #5: Yes

Reviewer #6: No

5. Review Comments to the Author

Reviewer #1: A cluster randomized controlled clinical trial was conducted to examine the impact of the FIFA 11+ warm-up on incidence and severity of injuries. The odds of sustaining injuries were lower in the intervention group compared to the control group.

Minor revisions:

1- Abstract: Provide the proportions that correspond to the number of sustained injuries in the following sentence. To improve clarity, replace “fewer” with “a lower proportion of.” “Fewer players sustained injuries in the intervention group (163 players) compared to the control group (200 players)...”

2- Double check that 20% is correct in the following statement. “A 20% reduction in the risk of sustaining moderate/severe injuries was observed in the intervention group (OR: 2.3, 95% C.L: 1.56-3.40).”

3- Paragraph beginning at line 117: The statistical power calculation should also indicate if the alpha level was one- or two-sided and statistical testing method which archives 90% power.

4- Line 124: Indicate the mechanism used to implement random allocation sequence (such as sequentially numbered containers), describe any steps taken to conceal the sequence until interventions were assigned.

5- Line 180: State the statistical method used to estimate the 95% confidence interval.

6- Table 1: The F and t values can be removed from Table 1. As a table footnote, indicate that p-values were calculated from t-test. Indicate if the distribution of the data was checked for normality prior to applying t-tests. Clarify the statistical method used to estimate the coefficient and its robust standard error.

7- Line 229: The statistical term for average is mean.

8- Table 2: Explain F-ratio variances and p-value variances.

9- Throughout the manuscript: Since the sample sizes of the control and intervention arms are not the same be sure to summarize the data as the proportion affected in addition to the frequency.

10- Table 3: Explain “Row %” and “Design Effect.”

11- Table 4: Consider removing columns of data for t-value and DF. Explain F-ratio variances, p-value variances, Mean Rank Intervention, Mean Rank Control, Sum of Ranks Intervention and Sum of Ranks Control. Possibly simplifying this table could improve clarity.

12- Table 5: Indicate if “Count” refers to sample size.

13- Indicate if any adverse events occurred during the course of the study.

14- Cite the statistical software used for the analysis.

15- An objective of the study was to address the severity of injuries; however, results for severity have not been presented.

Reviewer #2: Thank you for the opportunity to review this article, which bridges the gap between training theory/rehabilitation theory and its impact upon performance measures. Importantly, it addresses these issues as they apply to the topic of the FIFA 11+, which is an area of the sport science literature needing more attention.

I see this manuscript as generally well-written and clearly laid out. The authors have done a nice job of laying out the need for such a study with the Introduction, clearly describing the Methods and Results, and tying the present findings to related studies within the Discussion. Given the interventional nature of this study, in which the members of the research team trained the coaches of the intervention group, and then regularly traveled to conduct follow up meetings with the respective teams, a LOT of work went into this study.

My suggestions for improving this manuscript are largely related to “cleaning up” some English-usage issues, such as the omission of an article within some sentences, a few awkwardly written sentences that may present a lack of clarity for many readers, the lack of use of the Oxford comma, and so on.

In summary, the aim of this project has merit. I recommend the authors address the issues noted below prior to this manuscript appearing in this peer-reviewed journal. Thanks once again for the opportunity to review this paper.

Specific comments:

Line 21: suggest replacing “typically” for “mostly”

Line 26: suggest inserting “the” between “as” and “unit”

Line 60: the statement of “ranges from 18.8 to 1.2” does not seem correct to me. Is this a typo? Logic and conventional practice for stating ranges of numbers suggests these numbers may transposed, the second number may be missing a digit, etc.

Lines 63-64: Suggest revising as follows: “Injuries generally occur more frequently during matches…”

Line 85: Suggest revising as follows: “…was developed by the group of experts commissioned by FIFA to reduce perceived intrinsic risk factors…”

Line 92: Suggest adding a comma after “eccentric hamstring strength56,

Line 93: Suggest revising as follows: “…correct alignment of the hip, knee, and ankle joints during dynamic activity in young athletes.”

Line 95: Suggest revising as follows: “However, the positive impact of the ….”

Lines 128-129. This sentence is awkwardly written. Here is a suggested revision: The principal investigator, assisted by the Rwanda soccer technical director, governed the randomization process.

Line 131: It states “three” but 4 forms are listed. Revise for greater clarity.

Lines 156-157: Revise as follows for greater clarity: “…all the second division teams in the experimental group were invited…”

Line 159: While throughout this manuscript, the Oxford comma is not generally used, its absence here in particularly creates unnecessary confusion. Thus, I recommended revising as follows: “prevention, the development, and use…”

Line 165: Recommend revising: “corrections to flaws in technique.”

Line 191: Recommend revising: “…sessions were defined as medium compliance…”

Line 194: Recommend revising: “…were done for lower limb, such as location of ….”

Line 198: Add “the” after “in” and before “2016”

Line 209: Add a comma after “coaches”

Line 212: Revise to (Table 1).

Lines 224-225: It appears that the intervention group was the group that was significantly greater, but it is not clear as written. Revise this slightly, making this a bit more apparent to the typical reader.

Line 235: Add a comma after “control group”

Line 255: Revise to (Table 3).

Line 272: Revise to (Table 4).

Line 280: Revise as follows: “lost 40% less days; they exhibited a mean of …”

Line 282: Revise to (Table 5).

Line 295: Add comma after “training”. Again, use of the Oxford comma throughout this manuscript would help a bit for the overall presentation.

Line 296: Add a comma after “rates”

Line 297: Revise to: “rising to 91% prior to matches”.

Line 300: Add “previous” between “one” and “study”

Line 339: Add “previous” between “one” and “study”

Line 379: This is an incomplete sentence. Revise for greater clarity.

Line 396: This is another incomplete sentence. Revise for greater clarity.

Reviewer #3: Complements to the authors on this very nicely conducted trial. I have a number of comments regarding the statistical analysis and presentation of results, but the trial design is sound and the results seem quite optimistic and useful.

Comments:

1. Is there a typo in the introduction, paragraph one, first sentence: “… ranges from 18.8 to 1.2 per 1000 hours” or was it intentional to specify the range from high to low? I would expect to see the range specified from low to high, but I supposed either way is acceptable.

2. Specification of sample size in a cluster-randomized trial should address both the number of clusters and the number of participants per cluster. In other words, given that the number of clusters was fixed at 12 per arm, the sample size calculation should specify the number of individuals needed per cluster, given an assumed value for the intra-class correlation (ICC). The source and justification for the ICC, and the corresponding design effect, should be stated.

3. Statistical methods should be clearer about what methods are used to account for clustering. Logistic regression is not appropriate here given the large prevalence rates. Poisson regression is probably the best option for incidence rates. Linear regression for risk differences.

4. In the statistical analysis section, the software package used should be specified (in addition to the “Epi-info complex tables function”.)

5. In the “Characteristics…” section, group means for coach experience should be stated, along with the p-value; the U and Z statistics are not necessary in the text.

6. There is far too much information in Table 1. At the very most this table should show two columns (Intervention and Control Group means) and the p-value. The p-value column is not necessary, but OK to include it this journal requires it.

7. In the “Compliance…” section, the first sentence should start with “The teams in the intervention group…”, for clarity.

8. Delete the first sentence in the “Injuries” section about player-level injuries. This is summarized in the Effects of the intervention at the level of the individual players” section. Instead, overall incidence rates (Table 2) and relative risks should be summarized here.

9. Table 2 should be restructured to show the following information for overall/training/match injuries (three rows):

a. The numbers of injuries in each study arm

b. The numbers of exposure hours in each study arm

c. The incidence rate in each arm, and 95% confidence interval

d. The relative risk of injury comparing arms

e. The p-value

f. (Remove all other extraneous statistical content from this table)

10. Table 3 should be reduced substantially and take a similar format as Table 2, with just three rows (overall, training, match) and the following four columns:

a. Number of players with 1+ injuries in the intervention arm / number of participants in the intervention arm and percent: (n/N %)

b. Same for the control arm (n/N %)

c. The risk difference and 95% CI (Odds ratios are not appropriate with such high prevalence rates, and the relative risk is better calculated using incidence rates which is already done in Table 2)

d. P-value

e. (remove everything else from this table)

11. Revise the text in the “Effects of the intervention at the level of the players” section to provide a simple verbal summary of the new Table 3.

12. Revise Table 4 similarly by adding 95% CIs for the incidence rates and removing all statistical columns except the p-value which should be computed using clustered-date methods.

13. The section “Effect of the intervention on the severity of injuries” really combines two concepts: severity and impact on lost hours. Suggest separating them into:

a. Tabulation of injury severity categories in each study arm, counts and percentages (denominator for percentages being the total of the number of injuries in each arm.)

b. Comparison of days of lost play. (As in Table 5). Show means and ranges for this, not CIs, as there will be interest in the distribution of days lost, not just inference about the mean. And, define the mean: mean per player? per injured player? mean per team?

14. Once Tables 2-5 have been simplified and reformatted, it may be possible to combine them into one single, concise table of results.

15. Would be nice to include an appendix with more detail regarding the intervention. Nothing too long, e.g. just a page, but enough to give a reader a good sense for what is involved and how long it takes.

16. Discussion is generally good but it could be more concise and there are a few typos.

Reviewer #4: The study analyses the effect of a specific warm-up protocol for the prevention of injuries in soccer players. The manuscript is well written and the methodology of the study is appropriate according to the objective of the research. The main limitation of this paper is that it´s not clear whether this research adds anything new to the literature. The gaps in the literature need to be better described in order to justify what this paper adds to the field of knowledge.

Introduction:

It does not give a complete picture of what still needs to be studied about the effectiveness of the FIFA 11+ protocol. With a quick search you can find a systematic review of Baredo et al. (2014) where they include 12 articles analysing the effects of this protocol for injury prevention in soccer players, the narrative review of Bizzini and Dvorak (2015), or even a meta-analysis conducted by Thorborg et al. (2017). All these reviews conclude that the FIFA 11+ injury prevention programme is effective for soccer players of different levels and ages. Thus, the statement made by the authors at the end of the introduction “The positive impact of the FIFA 11+ warm up in preventing injuries has not yet been fully established, with poor compliance to the programme limiting conclusions drawn in several studies” needs to be clarified as it is difficult to sustain based on previous reviews. Without giving a complete picture of the state of the art it is not possible to determine how the submitted manuscript adds new knowledge or insights into this field of knowledge. The authors need better justify the need of this research.

The reference numbers are chaotic. In page 4 line 90 we go from the reference 29 to the number 507?? and next to the 56,…. Then in line 96 this numbers with no order at all are listed as references (30;17;31;23;24;20;32;33;25)

Methodology:

There is a lack of details about the intervention, the warm-up protocol. Although the programme is accessible for the FIFA web page it is not clear how exactly they implemented the warm-up protocol. For example the original programme includes three levels. Furthermore, the reference given for accessing the protocol does not lead to the programme itself (FIFA, F-Marc. The “11+” Manual: A complete warm-up programme to prevent injuries. 2007:1-76. http://www.yrsa.ca/pdf/Fifa11/11plus_workbook_e.pdf.) Thus, more details about the exact intervention are needed as it is key for practical applications and replication.

Discussion: similar to the introduction section, there is a need for better discussing the already existing literature about the effects of the FIFA 11+ programme for preventing injuries.

Reviewer #5: The authors have examining the effect of the FIFA 11+ injury prevention programme among male soccer players in an African population. This study showed that the FIFA 11+ programme with moderate to high compliance in training and match preparation decreased the incidence of injuries.

Line 60-61: ranges from 18.8 to 1.2 per 1000 hours.

Is here the lower number first or not?

Injuries occur

63-65: consistently more often during matches than during training and most commonly in the lower extremities

Is this proven?

Line 214 Players had around 3 years of experience in football? This is really small background. Please explain why only 3 years?

Were the coaches included in the analysis and if yes why?

Please explain the training programs of the control groups. Did they all had the same warm up?

Also explain the other parts of training plan during one week.

Reviewer #6: This manuscript explores the effects of a specific FIFA11+ warm-up intervention in the injury incidence in an important sample of amateur soccer players. In general terms, the manuscript is well organized, the research question is well defined and the results are clearly presented, although the use of English is far to reach the higher standards, and some concerns in methods section, as well as several statements in the discussion should be revised and qualified before more consideration for publication. Finally, authors make a correct use of references in their manuscript, but some references need to be reviewed according the journal's instructions. Below is shown some comments for the authors:

Major concerns:

Title: After reading the manuscript, I suggest that the term reduction should be changed by the more correct term incidence. The reduction is precisely an effect of the intervention.

Introduction.

Is the sample an amateur or profesional sample? I gather that is amateur, so theoretical framework should be more contextualized for amateur soccer players. I suggest that review several recent studies investigating specific warm-up protocols for injury prevention in amateur soccer players, like this study:

García-Luna, M.A.; Cortell-Tormo, J.M.; García-Jaén, M.; Ortega-Navarro, M.; Tortosa-Martínez, J. Acute Effects of ACL Injury-Prevention Warm-Up and Soccer-Specific Fatigue Protocol on Dynamic Knee Valgus in Youth Male Soccer Players. Int. J. Environ. Res. Public Health 2020, 17, 5608.

Methods.

Design and ethical approval: The experimental design and the randomization system of the teams should be explained in more detail in this section.

Participants: Please, add in this section the descriptive data of participants (number of players (total / per group), and mean and SD about age, experience, level of amateur/professionally should appear here in one sentence, apart from the Table 1. Otherwise, you should reference or insert there the Table 1 providing more info about demographic characteristics of players). Also, if randomization was based on teams instead of players, descriptive data of teams and their selection criteria should be explained. Lastly, which inclusion/exlusion criteria for player recruitment within respective teams was followed? What is the total dropout rate along the season? Explain more deitaled, please.

Sample size: This section should be renamed as "sample size calculation", and should specify the total number of selected players, and their distribution (number per group). The power analysis seems right.

Randomization and allocation of the participants: Why did you include the head coaches as participant in the study, if they were not subjects for intervention / control? On the other hand, explain more specifically the randomization criteria.

Instruments: Did you collected the coach's perceptions and/or opinions, or motivations, about different practical issues on the implementation of the FIFA11+ program as a warm-up after intervention? I think that some interviews to coaches could be collected, and a qualitative analysis could be performed. If you do not have these data, you should include this question as possible limitation and/or future research.

Intervention: I have some concern in this section: Firstly, there was no control on the usual warm up applied by the different control group coaches along the season. It can be understood that each one continued their usual warm up, but there is no information about this (what type of exercises, duration, intensity, of each warm up, and what frecuency of the sessions, etc. were performed?). I think that report these data would be necessary to check the heterogeneity -or not- of these different warm-up proposals, seeking for standardized systems, if you want to compare the different control warm-up effects with the experimental warm-up effects. Secondly, how it was controlled that coaches effectively applied the FIFA11+ program as a warm up thorough all trainings and matches of the seasons in each club? (if it was done, you have to include it in the section). Please, explain these regards.

Results:

Lines 198-212, and Figure 1: I think that this information about demographic characteristic of coaches and players should be located in the participant section. Also, I do not understand why the coaches were included as participant, if they did not performed the control or experimental warm-ups. I suggest exclude these data from analyses.

Lines 222-225: As it can be seen in this section, there was a significant difference on the total training exposure, that it could have influenced the injury incidence as external explaining factor of the injury incidence as well. You did not report the total exposure hours of control group (please provide it in this section), and other analysis comparing these data should be performed (if it was performed, please provide the significance p-value). If significant differences resulted, you should interpret your data considering this external factor as well.

Discussion:

Lines 292-301: You should specify that the experimental FIFA11+ program was applied as a warm up, and that the incidence of injuries and their severity decreased, always compared to the control programs (traditional programs?? You should comment this regard, as I was pointed above).

Lines 303-309: When you state about the generalization of results, you have to limite only to the amateur soccer players (or players with similar characteristic). Caution on the generalization of results.

Lines 344-346: Here, you stated: "he intervention resulted in less contact injuries, that can potentially be attributed to the neuromuscular effects of the FIFA 11+ programme". What is the basis to state this? I think that contact injury incidence has a multifactorial component that could not be explained by only one factor.

Lines 362-365: This hypothesis is an overstatement. You cannot attribute the effects of a specific warm-up to the recovery of different injuries. Please, qualify this. This should also be qualified in the Line 383, when you state that the experimental warm up reduced the contact injuries during matches (it seems an overstatement). Lines 390-391: Same concern.

Lines 375-379: The idea is not clear. Please, redact more clearly these sentences.

Compliance with the programme section: Similarly that methods section, how it was controlled that coaches effectively applied the FIFA11+ program as a warm up thorough all trainings and matches of the seasons in each club? Please, explain this essential question.

References: Please, review citations accordingly the journal instructions.

English review: I think that a native English speaker should review the English use throughout the manuscript.

**Minor concerns:

Introduction:

Lines 63-65: Authors talk here about the injury incidence. This sentence requires be referenced.

Line 97: this corresponds to the methods section (in study design).

Methods:

Line 108: Please, spell out the FERWAFA acronym, if it is the first time that appears in the manuscript.

Line 131: Instruments: you stated that three forms were used collecting data, but there is enumerated four different forms. Please, correct this regard.

Line 134: Is this form to control the player's exposure during training sessions? So, please specify this in the name of the form.

Line 138: You spell out FERWAFA here, but it appeared above, in line 108. Please, review this regard.

Lines 147 and 149: Please, provide reference about the definition proposed for injury, and the reference for classifying the severity of the injury, or explain the criteria followed to determine this

Discussion:

Line 292: Please, define RCT (it is the first time that appears in the manuscript).

Line 311: Please, specify that the initial training is referred to the pre-season instructional meeting. It is not clear here.

Lines 334-336: Please, provide references of previous studies that you comment here.

6. PLOS authors have the option to publish the peer review history of their article (what does this mean?). If published, this will include your full peer review and any attached files.

Reviewer #1: No

Reviewer #2: No

Reviewer #3: No

Reviewer #4: No

Reviewer #5: No

Reviewer #6: No

---

## [Author Response · Author response to Decision Letter 0]

4 Feb 2021

Response to the editor

Comment #1: Please ensure that your manuscript meets PLOS ONE's style requirements, including those for file naming. The PLOS ONE style templates can be found at

Response: 

Many thanks for your guidance. We have checked the style requirements and determined that our manuscript meets them.

Comment #2: Please include captions for your Supporting Information files at the end of your manuscript, and update any in-text citations to match accordingly. Please see our Supporting Information guidelines for more information: http://journals.plos.org/plosone/s/supporting-information

Response: 

We appreciate your comment. We updated accordingly. 

Response to reviewers

Reviewer #1

A cluster randomized controlled clinical trial was conducted to examine the impact of the FIFA 11+ warm-up on incidence and severity of injuries. The odds of sustaining injuries were lower in the intervention group compared to the control group.

Minor revisions:

Comment #1

Abstract: Provide the proportions that correspond to the number of sustained injuries in the following sentence. To improve clarity, replace “fewer” with “a lower proportion of.” “Fewer players sustained injuries in the intervention group (163 players) compared to the control group (200 players)...”

Response: Many thanks for the suggestion. The sentence is corrected and now reads “A lower proportion of players sustained injuries in the intervention group (52%) compared to the control group (63%)”. Similar wording has been changed throughout.

Comment #2

Double check that 20% is correct in the following statement. “A 20% reduction in the risk of sustaining moderate/severe injuries was observed in the intervention group (OR: 2.3, 95% C.L: 1.56-3.40).”

Response:

Thank you very much for the observation. The sentence was modified to suit the current updated results as suggested by other reviewers. 

Comment #3 

Paragraph beginning at line 117: The statistical power calculation should also indicate if the alpha level was one- or two-sided and statistical testing method which archives 90% power.

Response:

Many thanks for this suggestion, The information has been added and the text now reads: “Using this estimate and targeting a 20% reduction of injuries using power of 90% for generalised linear regression model and two-tailed alpha of 5%, a sample size of 125 was required per group.”

Comment #4

Line 124: Indicate the mechanism used to implement random allocation sequence (such as sequentially numbered containers), describe any steps taken to conceal the sequence until interventions were assigned.

Response:

Thank you very much for the suggestion. The following sentences were added for clarity. “The numbers representing the 24 teams were placed in sealed envelopes with an equal opportunity of being selected for groups. A research assistant selected the envelope for both groups drawing one by one to achieve random allocation by team to either the intervention group or the control group.”

Comment #5

Line 180: State the statistical method used to estimate the 95% confidence interval.

Response:

Thank you very much for the comment. The method used to calculate the 95% confidence intervals was inserted. 

Comment #6

Table 1: The F and t values can be removed from Table 1. As a table footnote, indicate that p-values were calculated from t-test. Indicate if the distribution of the data was checked for normality prior to applying t-tests. Clarify the statistical method used to estimate the coefficient and its robust standard error.

Response:

We appreciate your valuable observation. Following comment #18 from reviewer #2 and comment #6 from reviewer #3, the content was revised and simplified to remain with only mean and standard deviation from both the intervention and control group and their corresponding p-values. 

Comment #7

Line 229: The statistical term for average is mean.

Response:

Many thanks for your comment. We totally agree with you about the synonym of both terms. Following the structure of the sentence, we believe that the term average fits well instead of the term mean. 

Comment #8

Table 2: Explain F-ratio variances and p-value variances.

Response:

Many thanks for your comment. In line with the comment #9 from reviewer #3, the F-ratio variances and p-value variances were removed and the remaining variables were the number of injuries, exposure hours, and incidence (95% confidence intervals), with rate ratios and the p-values reported. 

Comment #9

Throughout the manuscript: Since the sample sizes of the control and intervention arms are not the same be sure to summarize the data as the proportion affected in addition to the frequency.

Response:

Many thanks for your valuable comment. The suggestion was taken into consideration, the frequency and proportion affected were presented especially in table 3. 

Comment #10

Table 3: Explain “Row %” and “Design Effect.”

Response:

We appreciate your comment. Table 3 was amended and combined with the results of table 4 as recommended by reviewer #3, comment #10, 11,14. 

Comment #11

Table 4: Consider removing columns of data for t-value and DF. Explain F-ratio variances, p-value variances, Mean Rank Intervention, Mean Rank Control, Sum of Ranks Intervention and Sum of Ranks Control. Possibly simplifying this table could improve clarity.

Response:

We appreciate your comment. It was addressed together with comment #10. 

Comment #12

Table 5: Indicate if “Count” refers to sample size.

Response:

Thank you for your observation. The content of the table was also revised and simplified. The results are presented according to the severity categories. The frequency and percentages are presented as well as the incidence and 95% confidence intervals. 

Comment #13

Indicate if any adverse events occurred during the course of the study.

Response:

Thank you very much for your observation. This information was provided in the last two sentences of the first paragraph of the discussion. 

Comment #14

Cite the statistical software used for the analysis.

Response:

Thank you for your comment. The information was inserted in the methods section under the statistical analyses subheading. 

Comment #15

An objective of the study was to address the severity of injuries; however, results for severity have not been presented.

Response:

Many thanks for your valuable comment. The results about the severity of injuries are now presented in table 4. 

Reviewer #2

Thank you for the opportunity to review this article, which bridges the gap between training theory/rehabilitation theory and its impact upon performance measures. Importantly, it addresses these issues as they apply to the topic of the FIFA 11+, which is an area of the sport science literature needing more attention.

I see this manuscript as generally well-written and clearly laid out. The authors have done a nice job of laying out the need for such a study with the Introduction, clearly describing the Methods and Results, and tying the present findings to related studies within the Discussion. Given the interventional nature of this study, in which the members of the research team trained the coaches of the intervention group, and then regularly traveled to conduct follow up meetings with the respective teams, a LOT of work went into this study.

My suggestions for improving this manuscript are largely related to “cleaning up” some English-usage issues, such as the omission of an article within some sentences, a few awkwardly written sentences that may present a lack of clarity for many readers, the lack of use of the Oxford comma, and so on.

Response:

Thank you for this suggestion, we have tried to clean up the sentence structure and hope that the clarity has improved as a result.

In summary, the aim of this project has merit. I recommend the authors address the issues noted below prior to this manuscript appearing in this peer-reviewed journal. Thanks once again for the opportunity to review this paper.

Thank you!

Specific comments:

Comment #1

Line 21: suggest replacing “typically” for “mostly”

Response:

We appreciate your valuable suggestion. We made the changes accordingly. 

Comment #2

Line 26: suggest inserting “the” between “as” and “unit”

Response:

We appreciate your comment. We made the changes as suggested. 

Comment #3

Line 60: the statement of “ranges from 18.8 to 1.2” does not seem correct to me. Is this a typo? Logic and conventional practice for stating ranges of numbers suggests these numbers may transposed, the second number may be missing a digit, etc.

Response:

Many thanks for the good observation. Another reviewer had also the same comment. We had actually written from highest to lowest. For clarity and understanding for the reader, the sentence now reads “… ranges from 1.2 to 18.8”

Comment #4

Lines 63-64: Suggest revising as follows: “Injuries generally occur more frequently during matches…”

Response:

Many thanks for the suggestion. We corrected it as advised. 

Comment #5

Line 85: Suggest revising as follows: “…was developed by the group of experts commissioned by FIFA to reduce perceived intrinsic risk factors…”

Response:

Many thanks for the suggestion. We inserted the content as suggested. 

Comment #6

Line 92: Suggest adding a comma after “eccentric hamstring strength56,

Response:

Many thanks for the observation. We inserted the comma as suggested. 

Comment #7

Line 93: Suggest revising as follows: “…correct alignment of the hip, knee, and ankle joints during dynamic activity in young athletes.”

Response:

Many thanks for the suggestion. We inserted the content as recommended. 

Comment #8

Line 95: Suggest revising as follows: “However, the positive impact of the ….”

Response:

We appreciate your comment. The content of the paragraph was revised and more content was added. 

Comment #9

Lines 128-129: This sentence is awkwardly written. Here is a suggested revision: The principal investigator, assisted by the Rwanda soccer technical director, governed the randomization process.

Response:

We appreciate your valuable suggestion. Small change suggested in the text: “The principal investigator, assisted by the Rwanda soccer technical director, supervised the randomization process.”

Comment #10

Line 131: It states “three” but 4 forms are listed. Revise for greater clarity.

Response:

Many thanks for your observation. We revised the title accordingly and it now reads: “Four forms were used to collect data:” 

Comment #11

Lines 156-157: Revise as follows for greater clarity: “…all the second division teams in the experimental group were invited…”

Response:

Many thanks for your suggestion. We inserted the content as suggested. 

Comment #12

Line 159: While throughout this manuscript, the Oxford comma is not generally used, its absence here in particularly creates unnecessary confusion. Thus, I recommended revising as follows: “prevention, the development, and use…”

Response:

Many thanks for your observation. We inserted the comma as suggested for more clarity. 

Comment #13

Line 165: Recommend revising: “corrections to flaws in technique.”

Response:

Many thanks for your suggestion. We inserted the content as suggested. 

Comment #14

Line 191: Recommend revising: “…sessions were defined as medium compliance…”

Response:

We appreciate your suggestion. We have adjusted this sentence to read: Compliance was computed based on each team’s implementation of the FIFA 11+ sessions throughout the season as a percentage of total training and match sessions and categorized as high compliance (> 75%), medium compliance (50% to 74%), or low compliance (<50%).

Comment #15

Line 194: Recommend revising: “…were done for lower limb, such as location of ….”

Response:

We appreciate your suggestion. Following your suggestions, it would be understood that the analyses of the location and the severity of injuries will only be for lower limbs; which is not the case in this study. It is the reason why we made other corrections to the sentence to read “Additional analyses were conducted for region of injury, lower limb injury location, and the severity of injuries. The categories of the severity of injuries were reflected by days of absence from full soccer participation45”

Comment #16

Line 198: Add “the” after “in” and before “2016”

Response:

We appreciate your suggestion. We made the corrections as suggested. 

Comment #17

Line 209: Add a comma after “coaches”

Response:

Based on the comments from other reviewers, the paragraph related to characteristics of the coaches was deemed not necessary and was therefore removed. 

Comment #18

Line 212: Revise to (Table 1).

Response:

Many thanks for your suggestion. Revision done.

Comment #19

Lines 224-225: It appears that the intervention group was the group that was significantly greater, but it is not clear as written. Revise this slightly, making this a bit more apparent to the typical reader.

Response:

Many thanks for your comment. The content was revised for clarity. 

Comment #20

Line 235: Add a comma after “control group”

Response:

Thank you very much your comment. We inserted the comma as suggested. 

Comment #21

Line 255: Revise to (Table 3).

Response:

Thank you for your suggestion. Revision done.

Comment #22

Line 272: Revise to (Table 4).

Response:

Thank you for your suggestion. Revision done.

Comment #23

Line 280: Revise as follows: “lost 40% less days; they exhibited a mean of …”

Response:

We appreciate your suggestion. We made the corrections as suggested. 

Comment #24

Line 282: Revise to (Table 5).

Response:

Thank you for your suggestion. Revision done.

Comment #25

Line 295: Add comma after “training”. Again, use of the Oxford comma throughout this manuscript would help a bit for the overall presentation.

Response:

We appreciate your suggestion. We made the corrections as suggested. 

Comment #26

Line 296: Add a comma after “rates”

Response:

We appreciate your suggestion. We made the corrections as suggested. The sentence is now reading “Along with significantly lower overall, training, and match incidence rates, the severity of injuries were lower in the intervention group and fewer days were lost due to injury.”

Comment #27

Line 297: Revise to: “rising to 91% prior to matches”.

Response:

Many thanks for your inputs. We made the corrections as suggested.

Comment #28

Line 300: Add “previous” between “one” and “study”

Response:

We appreciate your suggestion. We made the corrections as suggested. 

Comment #29

Line 339: Add “previous” between “one” and “study”

Response:

We appreciate your suggestion. We made the corrections as suggested. 

Comment #30

Line 379: This is an incomplete sentence. Revise for greater clarity.

Response:

Many thanks for your comment. The sentence has been revised. 

Comment #31

Line 396: This is another incomplete sentence. Revise for greater clarity.

Response:

Many thanks for your comment. We revised the sentence as recommended. 

Reviewer #3

Complements to the authors on this very nicely conducted trial. I have a number of comments regarding the statistical analysis and presentation of results, but the trial design is sound and the results seem quite optimistic and useful.

Thank you!

Comments:

Comment #1 

Is there a typo in the introduction, paragraph one, first sentence: “… ranges from 18.8 to 1.2 per 1000 hours” or was it intentional to specify the range from high to low? I would expect to see the range specified from low to high, but I supposed either way is acceptable.

Response:

Many thanks for the observation. Another reviewer had also the same comment. For clarity and understanding for the reader the sentence now reads: “… ranges from 1.2 to 18.8 per 100 hours”

Comment #2

Specification of sample size in a cluster-randomized trial should address both the number of clusters and the number of participants per cluster. In other words, given that the number of clusters was fixed at 12 per arm, the sample size calculation should specify the number of individuals needed per cluster, given an assumed value for the intra-class correlation (ICC). The source and justification for the ICC, and the corresponding design effect, should be stated.

Response:

We appreciate your comment. The design effect was mentioned and we added the value of ICC as advised. 

Comment #3

Statistical methods should be clearer about what methods are used to account for clustering. Logistic regression is not appropriate here given the large prevalence rates. Poisson regression is probably the best option for incidence rates. Linear regression for risk differences.

Response:

Thank you for your comment. The poisson regression analysis was used to analyse data instead of logistic regression. 

Comment #4. In the statistical analysis section, the software package used should be specified (in addition to the “Epi-info complex tables function”.)

Response:

Many thanks for this comment. The content of the other statistical software used in data analyses was inserted. “Microsoft Excel (Version 2000, Microsoft, Redmond, Wash.) was used to capture the data. Data analysis was carried out using “Statistica” data analysis software, version 13.2, Dell Inc. and the Statistical Package for Social Sciences (SPSS) for Windows, version 21.0, SPSS Inc, Chicago, USA. “ 

Comment #5. In the “Characteristics…” section, group means for coach experience should be stated, along with the p-value; the U and Z statistics are not necessary in the text.

Response:

Many thanks for the good observation. Based on a comment from reviewer#6 (comment#12) who deemed not necessary to include coaches’ data because they did not perform the warm-up exercises, we opted to exclude the presentation of data for the coaches and focus on the data related to the players. 

Comment #6. There is far too much information in Table 1. At the very most this table should show two columns (Intervention and Control Group means) and the p-value. The p-value column is not necessary, but OK to include it this journal requires it.

Response:

We appreciate your comment. The table was simplified as suggested. It contains only the mean (SD) values for the intervention and control group and one column for p-values. 

Comment #7

In the “Compliance…” section, the first sentence should start with “The teams in the intervention group…”, for clarity.

Response:

We appreciate your suggestion. We made the corrections as advised. 

Comment #8 

Delete the first sentence in the “Injuries” section about player-level injuries. This is summarized in the Effects of the intervention at the level of the individual players” section. Instead, overall incidence rates (Table 2) and relative risks should be summarized here.

Response:

We appreciate your comment. The first sentence in the “Injuries” section was deleted. The relevant information was also inserted as suggested. 

Comment #9 

Table 2 should be restructured to show the following information for overall/training/match injuries (three rows):

a. The numbers of injuries in each study arm

b. The numbers of exposure hours in each study arm

c. The incidence rate in each arm, and 95% confidence interval

d. The relative risk of injury comparing arms

e. The p-value

f. (Remove all other extraneous statistical content from this table)

Response:

Many thanks for your suggestion. We simplified the table 2 as recommended for simplicity and clarity. 

Comment #10 

Table 3 should be reduced substantially and take a similar format as Table 2, with just three rows (overall, training, match) and the following four columns:

a. Number of players with 1+ injuries in the intervention arm / number of participants in the intervention arm and percent: (n/N %)

b. Same for the control arm (n/N %)

c. The risk difference and 95% CI (Odds ratios are not appropriate with such high prevalence rates, and the relative risk is better calculated using incidence rates which is already done in Table 2)

d. P-value

e. (remove everything else from this table)

Response:

Many thanks for your suggestion. We simplified the content of table 3 and we combined the results with the content of table 4. 

Comment #11 

Revise the text in the “Effects of the intervention at the level of the players” section to provide a simple verbal summary of the new Table 3.

Response:

We appreciate your suggestion. We revised the content according to the results presented in table 3. 

Comment #12 

Revise Table 4 similarly by adding 95% CIs for the incidence rates and removing all statistical columns except the p-value which should be computed using clustered-date methods.

Response:

We appreciate your suggestion. The table was removed and its content was presented in table 3. 

Comment #13 The section “Effect of the intervention on the severity of injuries” really combines two concepts: severity and impact on lost hours. Suggest separating them into:

a. Tabulation of injury severity categories in each study arm, counts and percentages (denominator for percentages being the total of the number of injuries in each arm.)

b. Comparison of days of lost play. (As in Table 5). Show means and ranges for this, not CIs, as there will be interest in the distribution of days lost, not just inference about the mean. And, define the mean: mean per player? per injured player? mean per team?

Response:

Many thanks for your valuable comments. The table 5, now table 4, was reorganized to present the results according to the injury severity categories. We opted to reserve the results of the other severity measures in the text to avoid having many tables. 

Comment #14

Once Tables 2-5 have been simplified and reformatted, it may be possible to combine them into one single, concise table of results.

Response:

Many thanks for your insightful comment. We modified the contents and the design of the tables. We combined the content of table 3 and 4. 

Comment #15 

Would be nice to include an appendix with more detail regarding the intervention. Nothing too long, e.g. just a page, but enough to give a reader a good sense for what is involved and how long it takes.

Response:

We appreciate your comment. The content was submitted as a supporting document. 

Comment #16 

Discussion is generally good but it could be more concise and there are a few typos.

Response:

We appreciate your positive comment. We revised and corrected the errors as well as condensing some sentences for flow. 

Reviewer #4

The study analyses the effect of a specific warm-up protocol for the prevention of injuries in soccer players. The manuscript is well written and the methodology of the study is appropriate according to the objective of the research. The main limitation of this paper is that it´s not clear whether this research adds anything new to the literature. The gaps in the literature need to be better described in order to justify what this paper adds to the field of knowledge.

Comment #1

Introduction:

It does not give a complete picture of what still needs to be studied about the effectiveness of the FIFA 11+ protocol. With a quick search you can find a systematic review of Baredo et al. (2014) where they include 12 articles analysing the effects of this protocol for injury prevention in soccer players, the narrative review of Bizzini and Dvorak (2015), or even a meta-analysis conducted by Thorborg et al. (2017). All these reviews conclude that the FIFA 11+ injury prevention programme is effective for soccer players of different levels and ages. Thus, the statement made by the authors at the end of the introduction “The positive impact of the FIFA 11+ warm up in preventing injuries has not yet been fully established, with poor compliance to the programme limiting conclusions drawn in several studies” needs to be clarified as it is difficult to sustain based on previous reviews. Without giving a complete picture of the state of the art it is not possible to determine how the submitted manuscript adds new knowledge or insights into this field of knowledge. The authors need better justify the need of this research.

Response:

We appreciate your comment. The purpose of this study was to examine the effectiveness of the warm up programme in a different population group of youth amateurs engaging in competitive football. The setting in an African country where the FIFA 11+ was not used and in a league with relatively novice soccer players adds to the existing literature. We therefore updated the introduction section accordingly. 

Comment #2

The reference numbers are chaotic. In page 4 line 90 we go from the reference 29 to the number 507?? and next to the 56,…. Then in line 96 this numbers with no order at all are listed as references (30;17;31;23;24;20;32;33;25)

Response:

Many thanks for your comment. There references were revised.

Comment #3

Methodology:

There is a lack of details about the intervention, the warm-up protocol. Although the programme is accessible for the FIFA web page it is not clear how exactly they implemented the warm-up protocol. For example the original programme includes three levels. Furthermore, the reference given for accessing the protocol does not lead to the programme itself (FIFA, F-Marc. The “11+” Manual: A complete warm-up programme to prevent injuries. 2007:1-76. http://www.yrsa.ca/pdf/Fifa11/11plus_workbook_e.pdf.) Thus, more details about the exact intervention are needed as it is key for practical applications and replication.

Response:

Thank you for your comment. The detailed content was provided in the appendix (S3) submitted separately in the system. 

Comment #4

Discussion: similar to the introduction section, there is a need for better discussing the already existing literature about the effects of the FIFA 11+ programme for preventing injuries.

Response:

We appreciate your comment. We reviewed the discussion accordingly. 

Reviewer #5 

The authors have examining the effect of the FIFA 11+ injury prevention programme among male soccer players in an African population. This study showed that the FIFA 11+ programme with moderate to high compliance in training and match preparation decreased the incidence of injuries.

Comment #1

Line 60-61: ranges from 18.8 to 1.2 per 1000 hours.

Is here the lower number first or not?

Response:

Many thanks for the comment. Two other reviewers pointed this out as well. The sentence now reads, “… ranges from 1.2 to 18.8 per 100 hours”

Comment #2

63-65: Injuries occur consistently more often during matches than during training and most commonly in the lower extremities

Is this proven?

Response:

Many thanks for your observation. It is true that sometimes the occurrence of injuries might not be as described in the sentence above. However taking exposure time into consideration, the incidence of injuries was almost always reported to be higher during matches than training. In line with the suggestion of reviewer#2 comment #4, the sentence now read “Injuries generally occur more frequently during matches than during training and most commonly in the lower extremities”. 

Necessary references were inserted. 

Comment #3

Line 214 Players had around 3 years of experience in football? This is really small background. Please explain why only 3 years?

Response:

We appreciate your comment. It is true that the experience of the players in playing organized football with clear teams and regular training was around three years. In Rwanda, an African country where football development is not well organized, there are no football academies and structured youth competition except in schools. Many young people play just for fun and the love of the game. In 2004, there was a national programme to develop youth football. A second division league composed of players under the age of 20 years was initiated. We conducted the study when majority of players had the opportunity to play for about 3 years in organized teams. 

Comment #4

Were the coaches included in the analysis and if yes why?

Response:

Many thanks for your question. Coached were not included in the analysis. The information about the coaches that was initially presented in the analysis has been removed. 

Comment #5

Please explain the training programs of the control groups. Did they all had the same warm up?

Also explain the other parts of training plan during one week.

Response:

We appreciate your comment. We provided more explanation as suggested. Now the paragraph reads: ”Coaches of the teams in the control group were invited to attend a training on basic first aid and emergency procedures for one day. The coaches received training from the researcher on basic first aid and emergency procedures. The training covered topics relating to the occurrence of injuries, field assessments and phases of rehabilitation. First aid for injuries affecting specific body parts (ankle, knee, hip, groin, hamstring, hand and head) including concussion were also discussed. The discussion focused on specific conditions that affect players on the field such as nosebleeds, eye and dental injuries. They were requested to continue with the way they usually organise the warm up and practices for their players. There was no specific warm up regime in the control group. None of the control group coaches were familiar with the FIFA 11+ warm up programme. Typically the soccer training starts with a warm up, followed by technical and/or tactical drills. However, depending on the period of the season, physical training may include specific exercises for cardiovascular endurance, muscle strength, coordination and speed for performance.

Reviewer #6 

This manuscript explores the effects of a specific FIFA11+ warm-up intervention in the injury incidence in an important sample of amateur soccer players. In general terms, the manuscript is well organized, the research question is well defined and the results are clearly presented, although the use of English is far to reach the higher standards, and some concerns in methods section, as well as several statements in the discussion should be revised and qualified before more consideration for publication. 

Response:

Thank you, the manuscript has been reviewed for language, grammar and sentence structure, we hope that the paper reads clearly. We have responded to the specific comments below.

Finally, authors make a correct use of references in their manuscript, but some references need to be reviewed according the journal's instructions. Below is shown some comments for the authors:

Major concerns:

Comment #1

Title: After reading the manuscript, I suggest that the term reduction should be changed by the more correct term incidence. The reduction is precisely an effect of the intervention.

Response:

We appreciate your comment. Now the title reads: “Effect of the FIFA 11+ soccer specific warm up programme on the incidence of injuries: a Cluster-Randomised Controlled trial”

Introduction.

Comment #2

Is the sample an amateur or profesional sample? I gather that is amateur, so theoretical framework should be more contextualized for amateur soccer players. I suggest that review several recent studies investigating specific warm-up protocols for injury prevention in amateur soccer players, like this study:

García-Luna, M.A.; Cortell-Tormo, J.M.; García-Jaén, M.; Ortega-Navarro, M.; Tortosa-Martínez, J. Acute Effects of ACL Injury-Prevention Warm-Up and Soccer-Specific Fatigue Protocol on Dynamic Knee Valgus in Youth Male Soccer Players. Int. J. Environ. Res. Public Health 2020, 17, 5608.

Response:

Many thanks for your comment. It was taken onto consideration while updating the background of the study as reflected in the last paragraphs of the introduction section. 

Methods.

Comment #3

Design and ethical approval: The experimental design and the randomization system of the teams should be explained in more detail in this section.

Response:

We appreciate your comment. The overall design is included in this section. We felt that the description of the randomization process warranted a separate paragraph under the subheading “Randomisation and allocation of the participants” 

Comment #4

Participants: Please, add in this section the descriptive data of participants (number of players (total / per group), and mean and SD about age, experience, level of amateur/professionally should appear here in one sentence, apart from the Table 1. Otherwise, you should reference or insert there the Table 1 providing more info about demographic characteristics of players). 

Response:

Thank you very much for your comment. We added the content as requested. 

Comment #5

Also, if randomization was based on teams instead of players, descriptive data of teams and their selection criteria should be explained. 

Response:

Thank you for your comment. In line with revewer#1 comment #4, More content was added to provide more description on the randomization process of the teams. 

Comment #6

Lastly, which inclusion/exclusion criteria for player recruitment within respective teams was followed? What is the total dropout rate along the season? Explain more deitaled, please.

Response:

We appreciate your comment. We inserted a subheading and content describing the inclusion criteria of the players. In the introduction of the results section we mentioned the participants that dropped out and this was also reflected in the corresponding figure. 

Comment #7

Sample size: This section should be renamed as "sample size calculation", and should specify the total number of selected players, and their distribution (number per group). The power analysis seems right.

Response:

We appreciate your comment. We renamed the section as suggested. The information about the number of selected players from each group was provided in the introduction of the results section. 

Comment #8

Randomization and allocation of the participants: Why did you include the head coaches as participant in the study, if they were not subjects for intervention / control? On the other hand, explain more specifically the randomization criteria.

Response:

We appreciate your comment. In line with revewer#2 comment #9, we provided more explanations about the randomization process. 

Similar to the comment #4&5 from reviewer#5, the coaches were not included in the analysis. The information about the coaches that was initially presented in the methods and analysis sections was removed. 

Comment #9

Instruments: Did you collected the coach's perceptions and/or opinions, or motivations, about different practical issues on the implementation of the FIFA11+ program as a warm-up after intervention? I think that some interviews to coaches could be collected, and a qualitative analysis could be performed. If you do not have these data, you should include this question as possible limitation and/or future research. 

Response:

Many thanks for the observation. This study was part of the research project. The inquiry about coaches’ perception about injury prevention and the 11+ programme was gathered. We are preparing a separate manuscript on related specific results. We initially mentioned this in the discussion section under “compliance with the programme”. However, this information was also inserted in the limitations section. 

Comment #10

Intervention: I have some concern in this section: Firstly, there was no control on the usual warm up applied by the different control group coaches along the season. It can be understood that each one continued their usual warm up, but there is no information about this (what type of exercises, duration, intensity, of each warm up, and what frecuency of the sessions, etc. were performed?). I think that report these data would be necessary to check the heterogeneity -or not- of these different warm-up proposals, seeking for standardized systems, if you want to compare the different control warm-up effects with the experimental warm-up effects.

Response:

Many thanks for your comment. This was also noted by reviewer#5 comment#5. One of the advantages of the FIFA 11+ programme is the organization of multiple elements designed to address all components needed for optimal warm-up. The control group was not asked to change what they considered normal practice to provide a pragmatic comparison. This was especially relevant in this context where the soccer league is relatively new, players do not have as much experience and there is no current “control” protocol in the literature. While documenting the type of exercises, duration and intensity would be interesting, this would require different methods and aims, which were beyond the scope of this project.

We have provided more general explanation about the warm up conducted in the control group teams. Now the paragraph reads: ”… They were requested to continue with the way they usually organise the warm up and practices for their players. There was no specific warm up regime organised by the coaches in the control group. None of the control group coaches were familiar with the FIFA 11+ warm up programme. 

Typically the soccer training starts with a warm up, followed by technical and/or tactical drills. However, depending on the period of the season physical training may include specific exercises for cardiovascular endurance, muscle strength, coordination and speed for performance.” 

In addition we included the need for further research to assess the content of the control teams’ warm up content for comparison with the intervention teams. 

Comment #11

Secondly, how it was controlled that coaches effectively applied the FIFA11+ program as a warm up thorough all trainings and matches of the seasons in each club? (if it was done, you have to include it in the section). Please, explain these regards.

Response:

We appreciate your comment. We had initially planned to collect data pertaining to the use of the warm up programme each time it was conducted. However, coaches were did not always record the information. This was also reported in the limitations section. We therefore had to rely only on the frequency that they used the warm-up. This was also mentioned in the limitations sections. 

Results:

Comment #12

Lines 198-212, and Figure 1: I think that this information about demographic characteristic of coaches and players should be located in the participant section. Also, I do not understand why the coaches were included as participant, if they did not performed the control or experimental warm-ups. I suggest exclude these data from analyses.

Response: 

Many thanks for your valuable comment. We excluded the data for the coaches based on comments from multiple reviewers. Following the CONSORT checklist, participant demographics and the corresponding figure are strongly recommended to be presented at the beginning of the results section. 

Comment #13

Lines 222-225: As it can be seen in this section, there was a significant difference on the total training exposure, that it could have influenced the injury incidence as external explaining factor of the injury incidence as well. You did not report the total exposure hours of control group (please provide it in this section), and other analysis comparing these data should be performed (if it was performed, please provide the significance p-value). If significant differences resulted, you should interpret your data considering this external factor as well.

Response: 

Many thanks for your valuable comment. The total exposure hours for the control group were added. The comment is well noted and was taken into consideration in the revision of the discussion. 

Discussion:

Comment #14

Lines 292-301: You should specify that the experimental FIFA11+ program was applied as a warm up, and that the incidence of injuries and their severity decreased, always compared to the control programs (traditional programs?? You should comment this regard, as I was pointed above).

Response: 

Many thanks for your valuable comment. The content about the traditional/usual warm up was inserted.

Comment #15

Lines 303-309: When you state about the generalization of results, you have to limite only to the amateur soccer players (or players with similar characteristic). Caution on the generalization of results.

Response: 

Many thanks for your valuable comment. The discussion about the difference between the training exposure and its possible contribution to injury reduction was added. 

Comment #16

Lines 344-346: Here, you stated: "he intervention resulted in less contact injuries, that can potentially be attributed to the neuromuscular effects of the FIFA 11+ programme". What is the basis to state this? I think that contact injury incidence has a multifactorial component that could not be explained by only one factor.

Response: 

We appreciate your comment. The discussion about the multifactorial nature of contact injuries was inserted. 

Comment #17

Lines 362-365: This hypothesis is an overstatement. You cannot attribute the effects of a specific warm-up to the recovery of different injuries. Please, qualify this. This should also be qualified in the Line 383, when you state that the experimental warm up reduced the contact injuries during matches (it seems an overstatement). Lines 390-391: Same concern.

Response: 

We appreciate your comment. The referred sentences were removed for clarity.

Comment #18

Lines 375-379: The idea is not clear. Please, redact more clearly these sentences.

Response: 

Thank you for your comment. The sentence was revised for clarity.

Comment #19

Compliance with the programme section: Similarly that methods section, how it was controlled that coaches effectively applied the FIFA11+ program as a warm up thorough all trainings and matches of the seasons in each club? Please, explain this essential question.

Response: 

We appreciate your comment. Practically we could not fully supervise and control what the coaches are organizing and what the teams are doing on daily basis. However, we mentioned in the compliance section under the methodology: “The instructor and the principal investigator visited the intervention group coaches once a week for the first month of the season, followed by monthly visits to foster implementation and compliance. Additional instructions were provided where necessary.” In addition coaches submitted the records of the frequency of the implementation of the programme along with the exposure forms every two weeks. 

Comment #20

References: Please, review citations accordingly the journal instructions.

Response: 

Thank you very much for the comment. We revised the in-text references and updated the reference list accordingly.. 

Comment #21

English review: I think that a native English speaker should review the English use throughout the manuscript.

Response: 

Many thanks for the comment. Other than the first author, the other authors are all native English speakers. They have reviewed the manuscript and made changes to wording, sentence structure and grammar.

**Minor concerns:

Introduction:

Comment #22

Lines 63-65: Authors talk here about the injury incidence. This sentence requires be referenced.

Response: 

Many thanks for the comment. The references were inserted. 

Comment #23

Line 97: this corresponds to the methods section (in study design).

Response: 

Many thanks for the comment. The sentence was revised. 

Methods:

Comment #24

Line 108: Please, spell out the FERWAFA acronym, if it is the first time that appears in the manuscript.

Response: 

Many thanks for the comment. It was spelled out as recommended. 

Comment #25

Line 131: Instruments: you stated that three forms were used collecting data, but there is enumerated four different forms. Please, correct this regard.

Response: 

Many thanks for the comment. The subheading was corrected. 

Comment #26

Line 134: Is this form to control the player's exposure during training sessions? So, please specify this in the name of the form.

Response: 

Many thanks for the comment. The content was corrected. 

Comment #27

Line 138: You spell out FERWAFA here, but it appeared above, in line 108. Please, review this regard.

Response: 

Many thanks for the comment. The content was corrected accordingly. 

Comment #28

Lines 147 and 149: Please, provide reference about the definition proposed for injury, and the reference for classifying the severity of the injury, or explain the criteria followed to determine this

Response: 

Many thanks for the comment. The reference was inserted. 

Discussion:

Comment #29

Line 292: Please, define RCT (it is the first time that appears in the manuscript).

Response: 

Many thanks for the comment. The definition was provided. 

Comment #30

Line 311: Please, specify that the initial training is referred to the pre-season instructional meeting. It is not clear here.

Response: 

Many thanks for the comment. The section was reviewed for clarity. It now reads “The length of the training was three days, …”

Comment #31

Lines 334-336: Please, provide references of previous studies that you comment here.

Response: 

Many thanks for the comment. The references were inserted.

---

## [Decision Letter · Decision Letter 1]

5 May 2021

Effect of the FIFA 11+ soccer specific warm up programme on the incidence of injuries: a cluster-randomised controlled Trial

PONE-D-20-09354R1

Dear Dr. Nuhu,

We’re pleased to inform you that your manuscript has been judged scientifically suitable for publication and will be formally accepted for publication once it meets all outstanding technical requirements.

Kind regards,

Don Hoover

Guest Editor

PLOS ONE

Additional Editor Comments (optional):

See suggestion of Reviewer 1, which is to replace "average" with "mean" throughout the final version of the manuscript.

Reviewers' comments:

Reviewer's Responses to Questions

**Comments to the Author**

1. If the authors have adequately addressed your comments raised in a previous round of review and you feel that this manuscript is now acceptable for publication, you may indicate that here to bypass the “Comments to the Author” section, enter your conflict of interest statement in the “Confidential to Editor” section, and submit your "Accept" recommendation.

Reviewer #1: (No Response)

Reviewer #2: All comments have been addressed

Reviewer #4: All comments have been addressed

Reviewer #5: All comments have been addressed

2. Is the manuscript technically sound, and do the data support the conclusions?

Reviewer #1: Yes

Reviewer #2: Yes

Reviewer #4: Yes

Reviewer #5: Yes

3. Has the statistical analysis been performed appropriately and rigorously? 

Reviewer #1: Yes

Reviewer #2: Yes

Reviewer #4: Yes

Reviewer #5: Yes

4. Have the authors made all data underlying the findings in their manuscript fully available?

Reviewer #1: Yes

Reviewer #2: Yes

Reviewer #4: Yes

Reviewer #5: Yes

5. Is the manuscript presented in an intelligible fashion and written in standard English?

Reviewer #1: Yes

Reviewer #2: Yes

Reviewer #4: Yes

Reviewer #5: Yes

6. Review Comments to the Author

Reviewer #1: Throughout the manuscript: Replace the term "average" with "mean."

Reviewer #2: The authors have addressed all of my previous comments/suggestions. I personally see this as a valuable study which effectively assessed the efficacy of the FIFA 11+ regimen on a broad basis.

Reviewer #4: The authors have nicely addressed the questions raiesed by the reviewers. Only two minor details:

- Line 330 hadsignificantly (separate the words)

- In the the first reference, the title is in capital letters.

Reviewer #5: (No Response)

7. PLOS authors have the option to publish the peer review history of their article (what does this mean?). If published, this will include your full peer review and any attached files.

Reviewer #1: No

Reviewer #2: No

Reviewer #4: No

Reviewer #5: No

---

## [Editor Report · Acceptance letter]

14 May 2021

PONE-D-20-09354R1 

Effect of the FIFA 11+ soccer specific warm up programme on the incidence of injuries: a Cluster-Randomised Controlled trial 

Dear Dr. Nuhu:

I'm pleased to inform you that your manuscript has been deemed suitable for publication in PLOS ONE. Congratulations! Your manuscript is now with our production department. 

Kind regards, 

on behalf of

Dr. Don Hoover 

Guest Editor

PLOS ONE